# Polypharmacy in the Management of Arterial Hypertension—Friend or Foe?

**DOI:** 10.3390/medicina57121288

**Published:** 2021-11-23

**Authors:** Camelia Cristina Diaconu, Matei-Alexandru Cozma, Elena-Codruța Dobrică, Gina Gheorghe, Alexandra Jichitu, Vlad Alexandru Ionescu, Alina Crenguța Nicolae, Cristina Manuela Drăgoi, Mihnea-Alexandru Găman

**Affiliations:** 1Faculty of Medicine, “Carol Davila” University of Medicine and Pharmacy, 050474 Bucharest, Romania; matei.cozma@gmail.com (M.-A.C.); gheorghe_gina2000@yahoo.com (G.G.); mihneagaman@yahoo.com (M.-A.G.); 2Department of Internal Medicine, Clinical Emergency Hospital of Bucharest, 14461 Bucharest, Romania; 3Department of Gastroenterology, Colentina Clinical Hospital, 20125 Bucharest, Romania; 4Department of Pathophysiology, University of Medicine and Pharmacy of Craiova, 200349 Craiova, Romania; codrutadobrica@gmail.com; 5Department of Dermatology, “Elias” University Emergency Hospital, 011461 Bucharest, Romania; 6Department of Gastroenterology, Clinical Emergency Hospital of Bucharest, 14461 Bucharest, Romania; jichitualexandra@yahoo.com (A.J.); vlioax92@yahoo.com (V.A.I.); 7Department of Biochemistry, Faculty of Pharmacy, “Carol Davila” University of Medicine and Pharmacy, 020956 Bucharest, Romania; alina.nicolae@umfcd.ro (A.C.N.); cristina.dragoi@umfcd.ro (C.M.D.); 8Department of Hematology, Center of Hematology and Bone Marrow Transplantation, Fundeni Clinical Institute, 022328 Bucharest, Romania

**Keywords:** polypharmacy, hypertension, drug interactions, elderly, chronic diseases, multimorbidity

## Abstract

*Background and Objectives:* Polypharmacy is associated with drug–drug or food–drug interactions that may pose treatment difficulties. The objective of the study was to investigate the use of polypharmacy in hypertensive patients hospitalized in the Internal Medicine Clinic of a European referral hospital. *Materials and Methods:* We conducted a retrospective chart review study on patients identified by a database search of discharge diagnoses to assess the use of polypharmacy and identify potential drug-drug and food-drug interactions. *Results:* In total, 166 hypertensive patients (68.46 ± 12.70 years, range 42–94 years) were compared to 83 normotensive subjects (67.82 ± 14.47 years, range 22–94 years) who were hospitalized in the clinic during the same period. Polypharmacy was more common in hypertensive versus normotensive subjects (*p* = 0.007). There were no differences in terms of age, as well as major (0.44 ± 0.77 versus 0.37 ± 0.73 interactions/patient, *p* = 0.52) and minor (1.25 ± 1.50 versus 1.08 ± 1.84 interactions/patient, *p* = 0.46) drug–drug interactions between patients with and without hypertension. The mean number of drug–drug interactions (6.55 ± 5.82 versus 4.93 ± 5.59 interactions/patient, *p* = 0.03), moderate drug–drug interactions (4.94 ± 4.75 versus 3.54 ± 4.17, *p* = 0.02) and food–drug interactions (2.64 ± 1.29 versus 2.02 ± 1.73, *p* = 0.00) was higher in patients with hypertension versus their counterparts. *Conclusions:* The present study reinforces that polypharmacy is a serious concern in hypertensive patients, as reflected by the high number of potentially harmful drug–drug or food–drug interactions. We recorded higher numbers of comorbidities, prescribed drugs, and moderate drug–drug/food–drug interactions in hypertensive versus normotensive patients. A strategy to evaluate the number of discharge medications and reduce drug–drug interactions is essential for the safety of hypertensive patients.

## 1. Introduction

High blood pressure (BP), or arterial hypertension, i.e., systolic BP (sBP) ≥ 140 mmHg and/or diastolic BP (dBP) ≥ 90 mmHg, is the main modifiable cardiovascular risk factor for ischemic heart disease and stroke, end-stage kidney disease, premature death and disability, and, overall, for mortality of any cause [1,2,3,4,5,6,7,8]. Currently, high BP is considered a real global epidemic, with the latest estimates showing that around 1.13 billion people suffer from this disease globally, with its overall prevalence being around 30–45% [3,5,8]. In Romania, the prevalence of hypertension according to the largest epidemiological study conducted in our country, i.e., the Study for Evaluation of Prevalence of Hypertension and Cardiovascular Risk in an Adult Population in Romania (SEPHAR), is estimated at 40.1% [9].

The prevalence of hypertension increases with population aging, affecting more than half of all people aged 60–69 and more than three-fourths of all people over 70 years old [3,6,8]. Moreover, approximately 90% of people aged 55–65 years who do not have hypertension will develop hypertension by the age of 80–85 years [3]. In terms of sex, the prevalence of hypertension is generally higher in men (24%) than in women (20%). Even if men have higher BP values starting at a younger age, after the age of 60, women have higher BP values compared to men [1,3,5,6,7]. 

Ultimately, a worrying aspect regarding hypertension is the population’s awareness of this subject. Although, according to the latest estimates, approximately 59% of patients receive correct antihypertensive treatment (compared to 31% in previous years), only 34% have controlled BP (lower than 140/90 mmHg), and over 30% of patients are not aware that they have high BP [3]. Despite ongoing efforts to find new efficient drugs that lower BP, in addition to the classical drugs, such as diuretics, angiotensin-converting-enzyme (ACE) inhibitors, angiotensin-receptor blockers, beta-blockers, and calcium channel blockers, the number of people with high BP will increase by about 15–20% by 2050, reaching about 1.5 billion [3,7].

Polypharmacy is a complex phenomenon that has experienced an important increase in recent decades, determined mainly by the increase in life expectancy of the population and, at the same time, by a higher number of comorbidities that require the administration of a specific therapy [3,4]. Although the term polypharmacy appeared more than a century ago, an exact definition of polypharmacy has not yet been established. Numerous data in the literature put the threshold of five drugs per day as the limit beyond which the term polypharmacy can be used [10,11,12,13].

Multiple comorbidities, self-medication, multiple presentations in different settings for disorders requiring medication from different classes, and increased survival of diseases with high mortality in the past are all factors that have led to the prescription of a large number of drugs in people over 65 years old [14,15]. Although many experts point to the danger of polypharmacy, it is a growing phenomenon, estimating that in the next two decades, in the United States, there will be a doubling of both the number of elderly people and the number of drugs administered [16].

The effects of polypharmacy are numerous and should concern physicians and policy makers both in terms of finances (due to significant health costs) and medical consequences (due to multiple drug interactions) [17]. There is an increase in the number of drug interactions in elderly patients who receive medication prescribed by different doctors and who are diagnosed with conditions that may affect the pharmacokinetics of prescribed drugs (due to liver or kidney disease) [6]. At the same time, the interactions mentioned above can represent an additional risk factor for polypharmacy and can be interpreted as new pathologies that require additional medication, thus forming a vicious circle from which the patient suffers the most [9]. Moreover, the literature notes the correlations between polypharmacy and the risk of cognitive and functional impairment, which is considered a poor prognostic factor, due to the impact of drug interactions, prolonged hospitalizations, and the risk of decreased compliance to treatment [14,16,18].

In many instances, polypharmacy cannot be avoided, as many patients suffering from multiple chronic conditions will require drugs from different classes. This phenomenon has led to the birth of a new concept, namely that of “adequate polypharmacy” that defines a sufficient number of drugs to treat the existing pathologies and comorbidities of a patient [18]. Certain classes of drugs are associated with an increased risk of developing drug interactions, i.e., anticoagulants, antibiotics, psychiatric and antidiabetic medication, and last but not least, antihypertensive agents [10].

Thus, taking into account the fact that a large part of the Romanian population suffers from hypertension (>40% according with SEPHAR II cohort study) and that lifestyle changes do not accurately control BP values due to non-adherence, the treatment of hypertension using complex therapeutic schemes becomes a “necessary evil” [9]. Thus, the objective of this retrospective study was to evaluate the use of polypharmacy in patients diagnosed with primary hypertension who underwent treatment with various classes of antihypertensive agents, as well as to report the potential drug–drug and food–drug interactions in hypertensive subjects exposed to polypharmacy. To our knowledge, this is the first study of this type conducted on hypertensive patients from Romania. In addition, we aim to highlight the risk of polypharmacy in hypertensive patients with multiple comorbidities.

## 2. Materials and Methods

### 2.1. Setting and Participants

We conducted a retrospective chart review study on a sample of 249 patients, of whom 166 were diagnosed with primary hypertension and 83 were normotensive. The patients were hospitalized between January and February 2018 in the Internal Medicine Clinic of the Clinical Emergency Hospital of Bucharest, Romania, which is a referral emergency university hospital in Bucharest, the capital of Romania. The study sample was not calculated a priori by probabilistic methods, as we decided to include in our study all the patients who attended the clinic in January–February 2018 and had been diagnosed with primary hypertension (*n* = 166), whereas the subjects who did not suffer from hypertension were included in a comparison group (*n* = 83). As during these two months, the clinic registered a number of 275 hospitalizations, we excluded from the study a total of 26 subjects for the following reasons: The patients died during hospitalization, the patients were transferred to other clinics, the discharge treatment recommendations could not be retrieved in order to evaluate the use of polypharmacy/drug–drug/food–drug interactions, and the patients suffered from conditions such as dementia or psychiatric illnesses that made them unable to accurately give their consent.

The study was approved by the Ethics Committee of the Clinical Emergency Hospital of Bucharest, Romania (registration number 4263, approved on 8 May 2019), and was conducted based on the recommendations of the Declaration of Helsinki of 1975, as revised in 2008(5), and the national law.

### 2.2. Data Sources

The patients were identified by a database search of the electronic medical records of the Internal Medicine Clinic of the Clinical Emergency Hospital of Bucharest, Romania. The electronic discharge summaries, containing data on demographics, medical history, discharge diagnoses, and treatment recommendations, of the eligible patients were retrieved and an electronic database was computed using the variables of interest.

### 2.3. Variables of Interest

Patients diagnosed with primary hypertension were identified using the I10 International Classification of Diseases (ICD) code. The following variables were retrieved from the electronic discharge summaries: Age, sex, discharge diagnoses, diagnosis of primary hypertension (yes/no), hypertension grade, number of comorbidities, main comorbidities (other than hypertension), and number and names of drugs prescribed at discharge. The diagnosis of hypertension was based on the guidelines available in January–February 2018, i.e., the 2013 European Society of Hypertension (ESH)/European Society of Cardiology (ESC) guidelines for the management of arterial hypertension [19]. The assessment of the cardiovascular risk was based on the Systematic Coronary Risk Evaluation (SCORE) system [19].

Polypharmacy was defined as the prescription of five or more drugs per day in a single patient. To check for potential drug–drug and food–drug interactions, we employed a reliable and widely used online instrument, namely the Drug Interactions Checker [20]. For each patient, we entered the name of each drug prescribed one-by-one and then used the Check for Interactions function of the online instrument to generate an electronic Drug Interaction Report. The Drug Interactions Checker classifies drug–drug interactions into major (“the interaction possesses a significant clinical value and should be avoided since there are more risks versus benefits”), moderate (“the interaction possesses a moderate clinical significance and should usually be avoided or used only if necessary”), or minor (“the interaction possesses a minimal clinical significance, but in order to minimize any risks for the patient, an alternative drug should be considered, or the patient should be carefully monitored”). The online instrument also displays potential food–drug interactions, e.g., warfarin and grapefruit juice. The Drug Interactions Checker takes into consideration all types of interactions even if the patient received a polypill. For example, if the patient is prescribed metoprolol, furosemide, and spironolactone as three separate pills, it will signal that there are two moderate drug–drug interactions: furosemide–metoprolol and spironolactone–metoprolol. If the patient is prescribed metoprolol and a single pill with a fixed-dose combination of furosemide and spironolactone, the program displays the same two aforementioned drug–drug interactions: furosemide–metoprolol and spironolactone–metoprolol.

### 2.4. Statistics

Descriptive statistics (numbers and percentages) were used to analyze the data based on hypertension status, polypharmacy status, and to evaluate the prevalence of hypertension and polypharmacy, as well as the demographics. Analysis of variance (ANOVA), unpaired t-tests, chi-squared, and Mann–Whitney U tests, where appropriate, were employed to check the presence of associations. The statistical analysis of the data was performed at a 5% level of significance using Microsoft Excel (Microsoft Office Professional Plus 2013), MedCalc statistical software (MedCalc Software Ltd 2019, Ostend, Belgium) and QuickCalcs (GraphPad Software 2019, San Diego, CA, USA).

## 3. Results

The study group included 166 hypertensive (68.46 ± 12.70 years, range 42–94 years; 79 women (47.60%) and 87 men (52.40%)) and 83 normotensive patients (67.82 ± 14.47 years, range 22–94 years; 39 women (46.99%) and 44 men (53.01%)). Most hypertensive patients suffered from grade 3 (*n* = 72; 43.38%) hypertension, followed by grade 2 (*n* = 66; 39.76%) and grade 1 (*n* = 11; 6.62%) hypertension, as depicted in Table 1. In 10.24% of cases (*n* = 17), the hypertension grade was not mentioned in the discharge summary. In terms of total cardiovascular risk, most hypertensive subjects were assigned to the very-high-risk category (*n* = 63; 37.96%), as displayed in Table 2. A total of 50 subjects (30.12%) had moderate cardiovascular risk, 43 subjects had high risk (25.90%), and in 10 (6.02%) cases, the cardiovascular risk could not be calculated.

The most common comorbidities of the hypertensive patients were dyslipidemia (*n* = 70; 42.16%), chronic heart failure (HF) (*n* = 63; 39.37%), type 2 diabetes mellitus (T2DM) (*n* = 50; 30.12%), obesity (*n* = 44; 26.50%), coronary heart disease (CHD) (*n* = 41; 24.69%), and chronic kidney disease (CKD) (*n* = 29; 17.46%), as reported in Figure 1.

The most common antihypertensive agents employed in monotherapy or in drug combinations (either as single pills with fixed-dose or as multiple pills) in hypertensive subjects were diuretics (*n* = 126), beta-blockers (*n* = 94), ACE inhibitors (*n* = 82), calcium channel blockers (*n* = 67), and angiotensin receptor blockers (*n* = 46), as depicted in Table 3.

The majority of subjects received combinations of antihypertensive drugs (*n* = 132; 79.52%), whereas monotherapy was administered in only 20.48% (*n* = 34) of the patients diagnosed with hypertension. A total of 51 patients (30.72%) received combinations of at least two drugs but administered as separate pills: 38 patients received a combination of two (most commonly beta-blocker + ACE inhibitor, *n* = 17, or beta-blocker + angiotensin receptor blocker, *n* = 8), three (most commonly beta-blocker + angiotensin receptor blocker + diuretic, *n* = 4), or four (beta-blocker + angiotensin receptor blocker + calcium channel blocker + diuretic, *n* = 2) antihypertensive agents. A total of 81 (48.80%) patients received a combination of multiple antihypertensive agents but at least two of these drugs were prescribed as single pills with fixed-dose combinations. The most used single pills with fixed-dose combinations given were: Combinations of two diuretics (furosemide + spironolactone; *n* = 34), angiotensin receptor blocker + calcium channel blocker combinations (olmesartan + amlodipine, *n* = 15; candesartan + amlodipine, *n* = 6), ACE inhibitor + calcium channel blocker combinations (perindopril + amlodipine, *n* = 7), ACE inhibitor + diuretic combinations (perindopril + indapamide, *n* = 7), or ACE inhibitor + calcium channel blocker + diuretic combinations (perindopril + amlopidine + indapamide, *n* = 7). Of note, several patients received two single pills with fixed-dose combinations, e.g., furosemide + spironolactone and olmesartan + amlodipine. 

The main drug classes administered to hypertensive patients were statins (*n* = 89; 53.6%), antiplatelet agents (*n* = 69; 41.5%), proton pump inhibitors (*n* = 66; 39.7%), anticoagulants (*n* = 51; 30.7%), oral antidiabetics (*n* = 30; 18.0%), vitamins and minerals (*n* = 24; 14.4%), antianginal agents (*n* = 28; 16.8%), antibiotics (*n* = 21; 12.6%), and insulin (*n* = 7; 4.2%) (Table 4). In terms of anti-inflammatory and analgesic drugs, acetaminophen was prescribed in five patients (3.01%) and tramadol in three patients (1.80%). Nonsteroidal anti-inflammatory drugs (NSAIDs) were recommended to eight subjects (4.82%), the most common prescribed NSAIDs being naproxen (1.80%). 

The employment of polypharmacy was more common in hypertensive (*n* = 135; 81.33%) versus normotensive subjects (*n* = 54; 65.06%) (*p* = 0.007). There were no differences in terms of age (68.46 ± 12.70 versus 67.82 ± 14.56 years, *p* = 0.72), as well as major (0.44 ± 0.77 versus 0.37 ± 0.73 interactions/patient, *p* = 0.52) and minor (1.25 ± 1.50 versus 1.08 ± 1.84 interactions/patient, *p* = 0.46) drug–drug interactions between patients with and without hypertension. Patients with hypertension had more comorbidities (9.13 ± 3.52 versus 7.90 ± 3.82 comorbidities/patient, *p* = 0.01) and were prescribed more drugs (6.72 ± 2.58 versus 5.74 ± 3.18 drugs/day, *p* = 0.01) in comparison with normotensive controls. The mean number of drug–drug interactions (6.55 ± 5.82 versus 4.93 ± 5.59 interactions/patient, *p* = 0.03) and moderate drug–drug interactions (4.94 ± 4.75 versus 3.54 ± 4.17, *p* = 0.02), as well as food–drug interactions (2.64 ± 1.29 versus 2.02 ± 1.73, *p* = 0.00), was higher in patients diagnosed with hypertension versus their normotensive counterparts. Data regarding polypharmacy, drug–drug, and food–drug interactions are depicted in Table 5.

Antihypertensive agents were discovered to potentially lead to major drug–drug interactions (spironolactone–ramipril, spironolactone–candesartan, spironolactone–perindopril), moderate drug–drug interactions (metoprolol–spironolactone, metoprolol–furosemide, metoprolol–amlodipine), or minor drug–drug interactions (perindopril–amlodipine).

Several of the potential food–drug interactions in hypertensive patients were:-Amiodarone/atorvastatin/repaglinide–grapefruit/grapefruit juice (which can lead to an increase in the drug’s plasmatic concentration);-Metformin/rosiglitazone/sitagliptin–alcohol (risk of lactic acidosis);-Perindopril/valsartan/candesartan/ramipril–foods rich in potassium (risk of hyperkalemia);- Acenocoumarol–foods rich in vitamin K (in particular “beef liver, broccoli, Brussels sprouts, cabbage, lettuce, soy beans, spinach, watercress, and other green leafy vegetables”).

The age of the hypertensive patients correlated with their number of comorbidities (r = 0.395; *p* = 0.000), the mean number of drugs prescribed at discharge (r = 0.223; *p* = 0.003), and the mean number of drug–drug (r = 0.215; *p* = 0.005) and food–drug (r = 0.167; *p* = 0.031) interactions.

## 4. Discussion

Our study investigated the use of polypharmacy and the risk of drug–drug and food–drug interactions in a sample of patients diagnosed with primary arterial hypertension and managed in a reference hospital from Bucharest, Romania. Overall, the greatest proportion of the evaluated subjects suffered from grade 3 hypertension (~43%), had very high total cardiovascular risk (~38%), and had a high burden of cardiometabolic comorbidities (dyslipidemia, chronic HF, T2DM, obesity, and/or CHD were detected in ~25–42% of the study group). The individuals who suffered from hypertension received a combination of antihypertensive agents in ~80% of cases and were exposed to polypharmacy (≥5 drugs/day) in ~81% of cases. Both drug–drug and food–drug interactions were more frequently encountered in hypertensive versus normotensive patients.

### 4.1. Comorbidities Frequently Associated with Hypertension

Comorbidity is a common status that increases the complexity of healthcare. The association of two or more diseases is more common among the elderly. Barnett et al. evaluated the prevalence of comorbidities in high-income countries in older adults and reported a percentage higher than 50% for the association of two or three comorbidities and a percentage higher than 20% for more than three comorbidities [17]. According to the literature, the prevalence of comorbidities is higher among people diagnosed with hypertension, including subjects with normal BP [18]. These data are also supported by our study, which reported an average number of 9.13 ± 3.52 comorbidities per patient in the hypertensive patients group compared to 7.90 ± 3.82 comorbidities per patient in the normotensive group. Several comorbidities that are common among hypertensive subjects are obesity, impaired fasting glucose or T2DM, dyslipidemia, cardiovascular disease, CKD, liver disorders, thyroid disorders, anemia, and/or anxiety [15,18,19].

Our study supports the existing data in the literature. Thus, among the most common comorbidities found in hypertensive patients from the group of study were dyslipidemia, chronic HF, T2DM, obesity, CHD, and CKD.

Dyslipidemia has been shown to be more common in patients with high, uncontrolled BP [15,20,21]. In our study, the prevalence of dyslipidemia among patients diagnosed with hypertension was high (~42%). There are data that support a direct proportional relationship between BP values and serum lipid levels [21]. We did not directly follow this relationship, but given that patients with stage 3 (~43%) and stage 2 (~40%) hypertension dominated in our study, we can suppose an indirect relationship between the presence of dyslipidemia and high BP.

The relationship between arterial hypertension and chronic HF is well-known. High BP values progressively lead to left ventricular hypertrophy and chronic heart failure [22]. Compared to the general population, the relative risk of developing chronic HF among patients diagnosed with arterial hypertension has been shown to be 1.4 [23]. In our study, the percentage of hypertensive patients who had associated chronic HF was significant (~39%). Proper treatment of high BP can reduce the risk of developing chronic HF by about 50% [22].

Regarding the relationship between T2DM and arterial hypertension, high BP values are found in two-thirds of patients diagnosed with T2DM [24,25]. Our study validates the close relationship between high BP and T2DM. Thus, approximately one-third of hypertensive patients (~30%) had associated T2DM. The association of these two diseases increases the risk of other cardiovascular diseases [26]. Treatment of arterial hypertension in diabetic patients is more difficult, with resistant hypertension being more common among these patients. Usually, for the control of BP values in patients who have associated T2DM, it is necessary to combine two or more antihypertensive drugs [27].

Obesity, especially visceral fat distribution, is associated with hormonal, inflammatory, and endothelial wall changes, which contribute to high BP and increase cardiovascular mortality [28,29]. In our study, obesity was another common comorbidity among hypertensive patients, with a prevalence of 26.5%.

High BP is also often associated with CHD. In the Long-Term Intervention with Pravastatin in Ischaemic Disease (LIPID) study, the incidence of hypertension among patients with CHD was 41% [30]. We evaluated the inverse correlation and the prevalence of CHD among hypertensive patients, and we found that ~25% of hypertensive patients have associated CHD as a comorbidity. As mentioned above, increased BP values will cause left ventricular hypertrophy over time, with a reduction of the coronary reserve and an increase in myocardial oxygen demand. Finally, myocardial ischemia will appear. Hypertension induces endothelial dysfunction and contributes to the destabilization of atherosclerotic plaques [31].

The prevalence of hypertension among patients with CKD is high and increases with a decreasing glomerular filtration rate [32]. In our study, the prevalence of CKD among hypertensive patients was quite low compared to other comorbidities. Thus, less than one-third (~17%) of hypertensive patients have associated CKD. One of the mechanisms that lead to increased cardiovascular risk among patients with CKD is the appearance of arterial calcifications [33]. These patients need adjustment of the drug treatment according to the glomerular filtration rate, as reduced renal function leads to impairment of drug pharmacokinetics [32].

The association of comorbidities with hypertension has been shown to have a negative impact on patients’ quality of life, treatment efficacy, and also on healthcare costs [34]. For example, patients who have associated hypertension and anxiety disorders are less treatment-adherent, have a reduced quality of life, and their management imposes higher costs [34,35].

### 4.2. Polypharmacy in Patients with Hypertension

One of the problems of modern medicine is the therapeutic management of the phenomenon of multi-morbidity and the avoidance of polypharmacy. The prevalence of polypharmacy varies between 10% and 90% depending on several factors such as age, comorbidities, geographical area, or the actual definition used for polypharmacy [36]. In our study, ~81% of the patients with hypertension were at risk of developing harmful drug interactions, compared with ~65% of the patients with normal BP values. These results validate the data from the literature according to which arterial hypertension is associated with a high risk of polypharmacy.

The elderly, in particular, are not only likely to develop chronic conditions, but also to have multiple associated ailments, with a direct impact on their quality of life [37]. Multi-morbidity is a common phenomenon in the elderly, with important consequences on therapeutic management, explained by, on one hand, the impact that each disease can have on the evolution of another associated disease and, on the other hand, by the risk of polypharmacy [38]. There is currently no universally accepted definition for polypharmacy. In general, polypharmacy is considered when an individual uses ≥5 drugs and excessive polypharmacy is considered when the patient uses ≥10 drugs [10,18].

High BP is a condition that is often associated with multiple comorbidities and the risk of polypharmacy. A study that recruited 310 patients >65 years old reported that about 50% of patients who needed hospitalization received five or more medications [39]. In these patients, the phenomenon of polypharmacy was especially found in cardiovascular diseases (arterial hypertension, CHD, heart failure, and atrial fibrillation) [39]. In patients with hypertension, it is common to use several classes of drugs (diuretics, adrenergic inhibitors, beta-blockers, vasodilators) to control BP in order to reduce the risk of cardiovascular death. By comparing the results of our study with data from the literature, the classes of drugs used in hypertensive patients are similar: Diuretics, beta-blockers, ACE inhibitors, calcium channel blockers, and angiotensin receptor blockers. High BP is often associated with other conditions, which require other classes of drugs. All these factors significantly increase the risk of polypharmacy [40]. Medeiros dos Santos et al. demonstrated a common association between antihypertensive and lipid-lowering or antidiabetic drugs [41]. In their study, all polypharmacy cases involved at least one of these combinations [41]. In our study, in addition to lipid-lowering and antidiabetic drugs, antihypertensive medication was frequently associated with antiplatelet agents, proton pump inhibitors, anticoagulants, vitamins and minerals, antianginal agents, and antibiotics.

Some of the most important trials that have examined the therapeutic management and evolution of patients with hypertension are the Hypertension in the Very Elderly Trial (HYVET), Predictive Values of Blood Pressure and Arterial Stiffness in Institutionalized Very Aged Population (PARTAGE), and Systolic Hypertension in the Elderly Program (SHEP). HYVET included 3845 patients from Europe, Australia, China, and Tunisia, aged ≥ 80 years, with systolic BP > 160 mmHg [42]. This study concluded that the use of a thiazide diuretic associated or not with an ACE inhibitor resulted in a 30% reduction in stroke risk, 39% in fatal stroke, 21% in all-cause mortality, and 64% in heart failure [43]. The therapeutic target in this case was BP < 150/80 mmHg [43]. The PARTAGE trial included 1130 people aged ≥80 years from nursing homes in France and Italy [44]. This study showed an inversely proportional relationship between systolic BP < 130 mmHg, the use of more than two antihypertensive drugs, and the risk of death from all causes [44]. This raises the question of whether lowering BP or using several classes of antihypertensive drugs, or both, are factors that contribute to increased mortality in these patients [44]. The SHEP trial enrolled 4736 subjects aged ≥ 60 years and evaluated the effectiveness of antihypertensive treatment in reducing the risk of non-fatal or fatal stroke in patients with isolated systolic hypertension [45]. This study shows that lowering systolic BP to an average of 143 mmHg leads to a 36% decrease in the risk of non-fatal or fatal stroke, the risk of non-fatal or fatal myocardial infarction by 27%, and the risk of death of all causes by 13% [45]. All of these clinical trials used combinations of antihypertensive drugs to evaluate their effectiveness on morbidity and mortality. Unlike the HYVET and SHEP trials, which showed clear benefits of antihypertensive treatments, the PARTAGE trial discusses the negative impact of polypharmacy on the mortality of patients with high BP. Our study did not track the relationship between target BP values and patients’ morbidity and mortality. However, all hypertensive patients have been treated with a combination of several drug classes and the therapeutic outcomes may be influenced by drug interactions. Thus, ~31% of patients received combinations of at least two drugs, administered as separate pills, and ~49% of patients received a combination of multiple antihypertensive agents, with at least two of these drugs prescribed as single pills with fixed-dose combinations.

Polypharmacy has a negative impact, both clinically and economically. Thus, summarizing the data from the literature, polypharmacy leads to the following notable effects: Increased risk of using inappropriate medications, underuse of appropriate medications, harmful drug interactions, functional or cognitive decline, lower physical performance, more frequent side effects, increasing hospitalization rates because of adverse drug reactions, increasing short-term hospitalization rates, non-adherence to treatment, increased risk of frailty, and higher mortality [36,46].

Non-adherence to treatment may be due, on one hand, to the phenomenon of polypharmacy, and on the other hand, to old age. Thus, with the advancement in age, the risk of cognitive decline and its underlying consequences increases [47]. Among the risk factors for polypharmacy, age and the number of associated comorbidities play key roles. Our study demonstrates a directly proportional relationship between the age of the patients, number of comorbidities, number of drugs prescribed, and frequency of drug interactions.

### 4.3. Antihypertensive Drugs Interactions (Drug–Drug and Food–Drug Interactions)

*1. Drug–drug interactions.* Drug–drug interactions have emerged as a serious concern in polymedicated patients. Several therapeutic substances, as well as foods, can alter the bioavailability of drugs. Drug–drug interactions exert qualitative or quantitative changes in the effect of a certain drug by concomitant or successive administration of another substance [48]. This interaction might lead to alteration of the therapeutic effect or safety profile of one or both of the drugs. Either a change in the pharmacokinetics or pharmacodynamics might be responsible for this drug–drug interaction [48]. Given that the most common ailments associated with high BP are obesity, dyslipidemia, T2DM, impaired glucose tolerance, CHD, and CKD, the most common interactions occur between the antihypertensive medication and the medication associated with these pathologies. Our study supports these data, related to both the comorbidities associated with hypertension and the classes of drugs frequently combined in the treatment of hypertensive patients.

A review by Becker et al. reported that drug–drug interactions are responsible for a relatively small number of on-call presentations (0.054%), but among the elderly, this percentage is higher (4.8%) [49]. Another study showed that out of 350 drug prescriptions for ambulatory patients, 83.42% would have had possible drug–drug interactions, which is close to the percentage identified in our study, of 81.33% [50]. Kothari et al. also analyzed drug–drug interactions in polymedicated patients, discovering that 71.50% of the prescriptions had possible interactions [51]. Of these, the most common interactions were between atenolol and amlodipine, followed by metoprolol and amlodipine. Our study also found the combination between metoprolol and amlodipine as a potential cause of moderate drug–drug interaction. In addition, other major drug–drug interactions found in our study were spironolactone–ramipril, spironolactone–candesartan, and spironolactone–perindopril, moderate drug–drug interactions were metoprolol–spironolactone and metoprolol–furosemide, and minor drug–drug interactions were perindopril–amlodipine.

In our study, the prescribed number of drugs was 6.55 ± 5.82 per patient, a number approximately equal to that in the study of Subramanian et al. [47]. In this study, the most common drugs involved in drug–drug interactions were antihypertensive drugs, calcium supplements, nonsteroidal anti-inflammatory drugs (NSAIDs), antibiotics, and oral antidiabetics, whereas in our study, the most common associations were between antihypertensive drugs and statins, antiplatelet agents, proton pump inhibitors, and anticoagulants.

Studies show that the most common combinations of antihypertensive drugs responsible for adverse effects due to their interactions are atenolol–amlodipine, furosemide–enalapril, furosemide–telmisartan, and furosemide–atenolol [52]. Despite the fact that the Joint National Committee VII recommends the association of angiotensin II receptor antagonists with diuretics, beta-blockers with diuretics, and ACE inhibitors with diuretics, there are data that warn about the risk of hypotension with these associations. The causes of the interactions might be vasodilation and depleted intravascular volume, with these occurring contrary to their opposite effect on potassium levels [53]. 

Regarding the interactions between spironolactone and ACE inhibitors or angiotensin II receptor antagonists, which were mostly incriminated in our study for causing major interactions, studies have shown that certain conditions, such as a reduced glomerular filtration rate, advanced age, T2DM, or a dose of spironolactone >25 mg daily led to severe hyperkalemia when heart failure is also associated [54]. 

Other interactions revealed by our study were between spironolactone and metoprolol. Although recent research has shown the synergistic effect that metoprolol and spironolactone have on cardiac remodeling [55] and regional and systemic hemodynamics, caution should be taken in order to avoid hyperkaliemia [56,57].

Another drug that has possible interactions with antihypertensive medication is aspirin, an antiplatelet drug, which was used by 41.5% of the patients enrolled in our study. Pankti S. Patel et al. conducted a study on 350 patients regarding the potential adverse drug–drug interactions, revealing that aspirin was the most frequent drug involved. The most common interactions were seen when aspirin was combined with metoprolol [50]. 

Other classes of drugs responsible for possible side effects when associated with antihypertensive medications are antidiabetics (glyburide). ACE inhibitors are responsible for temporarily increasing the sensitivity to insulin, leading to a higher risk of hypoglycemia when associated with sulphonylureas [47,58]. In our study, 18.0% of patients were treated with oral antidiabetics and 4.2% were on insulin.

*2. Food–drug interactions.* Regarding the interactions of drugs with certain foods, side effects can occur by altering their absorption by diets with foods containing large amounts of fats, proteins, or fiber [59]. The cause of these interactions is the change in the bioavailability of the drugs, which correlates with their clinical effect. The most important interactions are caused by the chelation of drugs with certain components from food. Moreover, the body’s physiological response to food intake, and more precisely gastric acid secretion, can increase or decrease the bioavailability of certain drugs [60,61]. The most common causes of changes in the bioavailability of a drug are foods that are able to absorb certain medications that can act as chelators, alter the gastric pH, modify the intestinal motility, or are able to affect the transport of certain proteins [62].

Food intake also lowers the bioavailability of perindopril, an ACE inhibitor, by 35%. This is associated with the decrease in ACE inhibition that is clinically significant [61].

Grapefruits, Sicilian oranges, star fruits, and pomelos are the most frequently blamed for changing the bioavailability of drugs. Of these, grapefruit has the highest ability to interact with almost all types of drugs [63]. 

Studies have shown that propranolol, celiprolol, and felodipine’s bioavailability could be influenced by certain foods. Serum propranolol levels are elevated under conditions of a high protein diet [63]. Another beta-blocker’s absorption, celiprolol, is inhibited by the consumption of orange juice [64]. Grapefruit juice has also the ability to increase the bioavailability of felodipine [62,63].

Our study validates the possible interactions between grapefruit/ grapefruit juice and drugs such as amiodarone, atorvastatin, and repaglinide. The result of these interactions may be represented by an increase in the plasma concentrations of the drug. Other possible food–drug interactions highlighted by our study are metformin/rosiglitazone/sitagliptin–alcohol (risk of lactic acidosis); perindopril/valsartan/candesartan/ramipril–foods rich in potassium (risk of hyperkalemia); and acenocoumarol–foods rich in vitamin K (in particular “beef liver, broccoli, Brussels sprouts, cabbage, lettuce, soy beans, spinach, watercress, and other green leafy vegetables”).

A low-sodium diet is recommended for hypertensive patients. Licorice extract, a common component of dietary supplements, increases the levels of cortisol that reach the mineralocorticoid receptors, therefore leading to increased sodium retention and decreased serum potassium [65]. Thus, dietary supplements should be avoided while using diuretics, antihypertensive drugs that were also used in our study in 126 patients. Irrespective of their health ailments, patients should only consume dietary supplements (including vitamins and minerals) when prescribed by their attending physicians and should not self-medicate. In addition, clinical specialists should only prescribe supplements for proven vitamin/mineral deficiencies and on the basis of evidence-based medicine [66].

Our study has several strengths and limitations. To our knowledge, this is the first study to examine the presence of drug–drug and food–drug interactions in patients diagnosed with hypertension from Romania. Considering the high prevalence of this condition in our country, our results can be used to inform future research regarding the prescription of antihypertensive medication in individuals with hypertension and multimorbidity. In terms of limitations, we must acknowledge that the study sample employed was small and thus our data may not reflect the risk of polypharmacy at a national level. Moreover, the assessment had a retrospective design and thus we were not able to check the adherence of the subjects to their prescribed medication. However, our research clearly points out that strategies to reduce the use of polypharmacy in hypertensive patients are warranted and that the management of hypertension is complex and would also require the insight of clinical pharmacists/pharmacologists, as well as dieticians and other health professionals [67,68,69]. In addition, as telemedicine has emerged as an important tool in the management of chronic disorders, in the near future, digital medication information and automated drug interaction analysis systems might become more popular among hypertension specialists and their patients [67,69]. All in all, further research, preferably with a prospective design and recruiting a larger patient sample, is needed to investigate the issues associated with the use of polypharmacy in subjects living with hypertension and to propose solutions to these clinical problems.

## 5. Conclusions

The present study reinforces polypharmacy as a serious concern in hypertensive patients, as reflected by the high number of potentially harmful drug–drug or food–drug interactions. We recorded higher numbers of comorbidities, prescribed drugs, and moderate drug–drug/food–drug in hypertensive patients vs. controls. A strategy to evaluate the number of discharge medications and reduce drug–drug interactions is essential for the safety of hypertensive patients. 

## Figures and Tables

**Figure 1 medicina-57-01288-f001:**
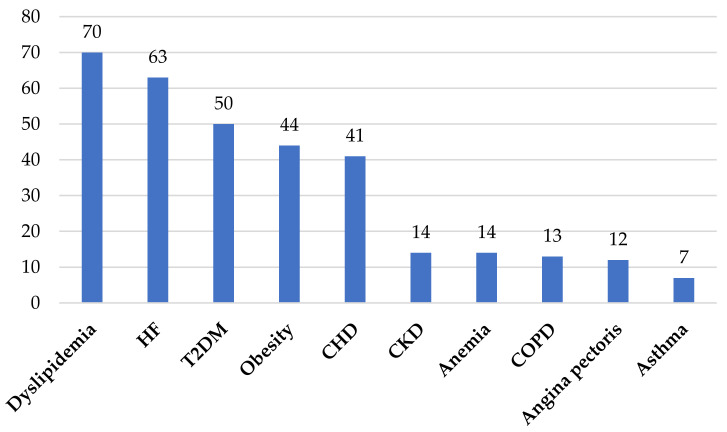
The most common comorbidities that coexisted with hypertension in the study group. *Legend:* COPD, chronic obstructive pulmonary disease. CKD, chronic kidney disease. CHD, coronary heart disease. T2DM, type 2 diabetes mellitus. HF, chronic heart failure.

**Table 1 medicina-57-01288-t001:** Distribution of the study group by hypertension stage.

Drug Class	Percentage (%)	Number of Patients
Stage 1	6.62%	11
Stage 2	39.76%	66
Stage 3	43.38%	72
Unspecified	10.24%	17
Total	100.00%	166

**Table 2 medicina-57-01288-t002:** Distribution of the study group in terms of total cardiovascular risk.

Drug Class	Percentage (%)	Number of Patients
Intermediate	30.12%	50
High	25.90%	43
Very high	37.96%	63
Unspecified	6.02%	10
Total	100.00%	166

**Table 3 medicina-57-01288-t003:** Antihypertensive agents employed in the management of hypertension in our study.

Drug Class	Type of Drug	Number of Patients
Diuretics (*n* = 126)	Furosemide	47
	Indapamide	38
	Spironolactone	36
	Hydrochlorothiazide	5
Beta-blockers (*n* = 94)	Metoprolol	45
	Nebivolol	21
	Carvedilol	14
	Bisoprolol	14
Angiotensin-converting-enzyme (ACE) inhibitors (*n* = 82)	Perindopril	57
	Ramipril	14
	Enalapril	5
	Captopril	2
	Lisinopril	2
	Trandolapril	2
Calcium channel blockers (*n* = 67)	Amlodipine	50
	Diltiazem	8
	Lercanidipine	5
	Felodipine	2
	Verapamil	1
	Nifedipine	1
Angiotensin receptor blockers (*n* = 46)	Candesartan	22
	Olmesartan	16
	Irbersartan	4
	Valsartan	3
	Telmisartan	1

**Table 4 medicina-57-01288-t004:** Main drug classes prescribed to hypertensive patients.

Drug Class	Number	Percentage
Statins	89	53.6%
Antiplatelet agents	69	41.5%
Proton pump inhibitors	66	39.7%
Anticoagulants	51	30.7%
Oral antidiabetics	30	18.0%
Antianginal agents	28	16.8%
Vitamins and minerals	24	14.4%
Antibiotics	21	12.6%
Insulin	7	4.2%

**Table 5 medicina-57-01288-t005:** Polypharmacy, drug–drug, and food–drug interactions in our study.

Parameter [per Patient]	Hypertensive	Normotensive	*p*-Value
Age [years]	68.46 ± 12.70	67.82 ± 14.56	0.72
Comorbidities	9.13 ± 3.52	7.90 ± 3.82	0.01
Prescribed drugs	6.72 ± 2.58	5.74 ± 3.18	0.01
Drug–drug interactions	6.55 ± 5.82	4.93± 5.59	0.03
Minor drug–drug interactions	1.25 ± 1.50	1.08 ±1.84	0.46
Moderate drug–drug interactions	4.94 ± 4.75	3.54 ± 4.17	0.02
Major drug–drug interactions	0.44 ± 0.77	0.37 ±0.73	0.52
Food–drug interactions	2.64 ± 1.29	2.02 ± 1.73	0.00

## Data Availability

The data presented in this study are available on request from the corresponding author.

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
