# Peer review of "Polypharmacy in the Management of Arterial Hypertension—Friend or Foe?"

_medicina, 2021, doi:10.3390/medicina57121288_

Round 1

Reviewer 1 Report

Polypharmacy is a common public health problem particularly among the patients with cardiovascular diseases associated with increased number of comorbidities in older age. The authors clearly stated that the intention of doctors to prescribe appropriate medication to their patients is usually challenged with the need to prescribe several different drugs at the same time.  However, the increased number of prescribed drugs will certainly lead to an increased health problems. These include adverse drug reactions and
potentially harmful drug–drug interactions, increased
exposure to potentially inappropriate medicines, poor
adherence and reducing the health benefits of prescribed
medications. 

The manuscript entitled "Polypharmacy in the Management of Arterial Hypertension – Friend or Foe?" gives us the insight on prescribing patterns  in the Internal Medicine Clinic of the Clinical Emergency Hospital of Bucharest, Romania.

Although the study is based on limited number of patients (n=249; "of 
whom 166 were diagnosed with primary hypertension and 83 were normotensive"), and which was monitored only for two months, the results are quite interesting and conclusive. 

Major comments

Introduction

  • Line 71, 72.  The authors stated that the "classical drugs for hypertension treatment are: diuretics, angiotensin-converting-enzyme (ACE) inhibitors, angiotensin-receptor blockers and calcium channel blocker". 

Comment: Add beta-blockers, since this class of drugs will appear in results and will be elaborated in discussion as well. 

Results

  • It is not so common that data in scientific journals are presented by pie-charts (Figure 1 and 2). This type of data presentation is more appropriate for PowerPoint presentations, if any. So, I recommend authors to redesign the figures 1 and 2 by using the bars or, even better to present the data in a table rather than pie-chart.
  • Figure 3. It would be more appropriate to show the data using the bars in descending order (from higher to lower incidence). 
  • The data presented in Table 1. clearly showed that authors used in consideration the monocomponent drugs only. What about the number of patients whom the fixed drug combinations were prescribed (like ACEI with thiazide diuretics and other combinations with indapamid and others)? It is not likely that cardiologists from the Clinical Emergency Hospital in Bucharest do not prescribe the fixed combinations of antihypertensives in their daily routine practice. 
  • Among the main drug classes administered to hypertensive patients there are no drugs belonging to NSAID group? It is unlikely that among the hypertensive patients there are no patients with arthritis or those needing analgesics and anti-inflammatory drugs. It is well known that NSAID are responsible not only for diminishing the antihypertensive effects of classical antihypertensives but also can induce very serious side effects when used in combination, particularly with ACEI and thiazides. 

Discussion

  • Lines 261-270. and Lines 376-387. The authors used bullets for listing the comorbidities and polypharmacy negative impacts???.

Commnet: This type of listing is more appropriate for 'Students textbooks' rather than for scientific journals. So, my honest recommendation to authors to put it in one sentence. 

  • The very last paragraph is considering the "Recommendations regarding the avoidance of polypharmacy in hypertensive patients"?? This type of "recommendations" are more suitable for Clinical guidelines rather than for research papers.  So, after discussing the main findings of their research and emphasizing the main limitations of their study, the authors should finish the paper with appropriate conclusions, like they actually did in the last paragraph. 
  • The sentences related to 'limitations of the study' are missing and should be added to the text.

Author Response

Dear Academic Editor,

Dear Peer-Reviewers,

We are very thankful to you for the pertinent notes; we have carefully read the comments and have revised/ completed the manuscript accordingly. Our responses are given in a point-by-point manner below, as well, all the changes to the manuscript are highlighted in yellow.

We hope that in this new form, the manuscript will be suitable for publication in Medicina.

Reviewer 1

Polypharmacy is a common public health problem particularly among the patients with cardiovascular diseases associated with increased number of comorbidities in older age. The authors clearly stated that the intention of doctors to prescribe appropriate medication to their patients is usually challenged with the need to prescribe several different drugs at the same time. However, the increased number of prescribed drugs will certainly lead to an increased health problems. These include adverse drug reactions and
potentially harmful drug–drug interactions, increased
exposure to potentially inappropriate medicines, poor
adherence and reducing the health benefits of prescribed
medications.  The manuscript entitled "Polypharmacy in the Management of Arterial Hypertension – Friend or Foe?" gives us the insight on prescribing patterns in the Internal Medicine Clinic of the Clinical Emergency Hospital of Bucharest, Romania.

Although the study is based on limited number of patients (n=249; "of 
whom 166 were diagnosed with primary hypertension and 83 were normotensive"), and which was monitored only for two months, the results are quite interesting and conclusive. 

We would like to thank for your valuable comments which helped us improve the manuscript. All suggestions were taken into consideration and appropriate information as well as required corrections were provided. New/corrected parts are highlighted in yellow to facilitate the assessment of changes. We did our best to fulfil the expectations and we hope that you will be satisfied with our corrections.

Major comments

Introduction

  • Line 71, 72.  The authors stated that the "classical drugs for hypertension treatment are: diuretics, angiotensin-converting-enzyme (ACE) inhibitors, angiotensin-receptor blockers and calcium channel blocker". 

Comment: Add beta-blockers, since this class of drugs will appear in results and will be elaborated in discussion as well. 

Response: Thank you for this valuable suggestion. We apologize for having forgotten to mention beta-blockers. We have revised the phrase as follows: “in addition to the classical drugs such as diuretics, angiotensin-converting-enzyme (ACE) inhibitors, angiotensin-receptor blockers, beta-blockers and calcium channel blockers“.

Results

  • It is not so common that data in scientific journals are presented by pie-charts (Figure 1 and 2). This type of data presentation is more appropriate for PowerPoint presentations, if any. So, I recommend authors to redesign the figures 1 and 2 by using the bars or, even better to present the data in a table rather than pie-chart.

Response: Thank you for this valuable suggestion. We agree with you that pie-charts are more suitable for PowerPoint presentations. We have replaced Figure 1 and Figure 2 with two Tables (Table 1 and Table 2) as suggested.

  • Figure 3. It would be more appropriate to show the data using the bars in descending order (from higher to lower incidence). 

Response: Thank you for this valuable suggestion. We agree with your comment. We have redesigned Figure 3 to display the comorbidities in descending order (from higher to lower incidence).

  • The data presented in Table 1. clearly showed that authors used in consideration the monocomponent drugs only. What about the number of patients whom the fixed drug combinations were prescribed (like ACEI with thiazide diuretics and other combinations with indapamid and others)? It is not likely that cardiologists from the Clinical Emergency Hospital in Bucharest do not prescribe the fixed combinations of antihypertensives in their daily routine practice. 

Response: Thank you for this valuable observation. In fact, most patients were prescribed combinations of antihypertensive drugs (either as multiple pills or in the form of polypills, i.e., single pills with fixed-dose combinations). We have added a separate paragraph to clarify this issue. In the first version of the paper, we did not present these data because due to the numerous combinations of drugs (>30 combinations) we considered it confusing. In fact, the drug-drug and food-drug interactions checker takes into consideration all types of interactions even if the patient received a polypill. For example, if the patient is prescribed metoprolol, furosemide and spironolactone as three separate pills, it will signal that there are two moderate drug-drug interactions: furosemide – metoprolol and spironolactone – metoprolol. If the patient is prescribed metoprolol and a single pill with a fixed-dose combination between furosemide and spironolactone, the program displays the same two aforementioned drug-drug interactions: furosemide – metoprolol and spironolactone – metoprolol.  

We have completed the manuscript with the following sentences: The most common antihypertensive agents employed in monotherapy or in drug combinations (either as single pills with fixed-dose or as multiple pills) in hypertensive subjects were diuretics (n=126), beta-blockers (n=94), ACE inhibitors (n=82), calcium channel blockers (n=67) and angiotensin receptor blockers (n=46), as depicted in Table 3.

The majority of subjects received combinations of antihypertensive drugs (n=132; 79.52%), whereas monotherapy was administered in only 20.48% (n=34) of the patients diagnosed with hypertension. A total of 51 patients (30.72%) received combinations of at least two drugs but administered as separate pills: 38 patients received a combination of two (most commonly beta-blocker + ACE inhibitor, n=17, or beta-blocker + angiotensin receptor blocker, n=8), three (most commonly beta-blocker + angiotensin receptor blocker + diuretic, n=4) or four (beta-blocker + angiotensin receptor blocker + calcium channel blocker + diuretic, n=2) antihypertensive agents. A total of 81 (48.80%) patients received a combination of multiple antihypertensive agents but at least two of these drugs were prescribed as single pills with fixed-dose combinations. The most used single pills with fixed-dose combinations given were: combinations of two diuretics (furosemide + spironolactone; n=34), angiotensin receptor blocker + calcium channel blocker combinations (olmesartan + amlodipine, n=15; candesartan + amlodipine, n=6), ACE inhibitor + calcium channel blocker combinations (perindopril + amlodipine, n=7), ACE inhibitor + diuretic combinations (perindopril + indapamide, n=7) or ACE inhibitor + calcium channel blocker + diuretic combinations (perindopril + amlopidine + indapamide, n=7). Of note, several patients two single pills with fixed-dose combinations, e.g., furosemide + spironolactone and olmesartan + amlodipine.

  • Among the main drug classes administered to hypertensive patients there are no drugs belonging to NSAID group? It is unlikely that among the hypertensive patients there are no patients with arthritis or those needing analgesics and anti-inflammatory drugs. It is well known that NSAID are responsible not only for diminishing the antihypertensive effects of classical antihypertensives but also can induce very serious side effects when used in combination, particularly with ACEI and thiazides. 

Response: Thank you for this valuable comment. NSAIDs/analgesics/opioids were prescribed in a low number of patients. Physicians are probably well aware that these drugs interact with anti-hypertensive drugs and probably this is the reason why these pharmacological agents were not prescribed at discharge. We have completed the manuscript’s Results section with the following phrase: “In terms of anti-inflammatory and analgesic drugs, acetaminophen was prescribed in 5 patients (3.01%) and tramadol in 3 patients (1.80%). Nonsteroidal anti-inflammatory drugs (NSAIDs) were recommended to 8 subjects (4.82%), the most common prescribed NSAIDs being naproxen (1.80%). “.

Discussion

  • Lines 261-270. and Lines 376-387. The authors used bullets for listing the comorbidities and polypharmacy negative impacts???.

Comment: This type of listing is more appropriate for 'Students textbooks' rather than for scientific journals. So, my honest recommendation to authors to put it in one sentence. 

Response: Thank you for this valuable suggestion. We synthesized this information in the form of a sentence.

  • The very last paragraph is considering the "Recommendations regarding the avoidance of polypharmacy in hypertensive patients"?? This type of "recommendations" are more suitable for Clinical guidelines rather than for research papers.  So, after discussing the main findings of their research and emphasizing the main limitations of their study, the authors should finish the paper with appropriate conclusions, like they actually did in the last paragraph. 

Response: Thank you for this valuable suggestion. We have removed these recommendations, discussed the limitations of our study and the conclusions.

  • The sentences related to 'limitations of the study' are missing and should be added to the text.

Response: Thank you for this valuable suggestion. We added a separate paragraph for strengths and limitations before the Conclusions.

Thank you for taking time to review our paper and for your valuable suggestions. Overall, we believe that, thanks to you pertinent comments, the quality of the manuscript has been improved and its revised version warrants publication in Medicina.

Reviewer 2 Report

At the present time, polypragmasy is a serious public health problem, as it is clinically manifested by a reduction in the effectiveness of pharmacotherapy, by the development of severe adverse drug reactions, and by a considerable increase in healthcare expenditures. The reason for the simultaneous prescription of multiple drugs may be comorbidity (multimorbidity), the availability of drugs, as well as  treatment standards that contain recommendations for using combination therapy with more than 5 drugs for only one disease in some cases, the efficiency of which corresponds to a high level of evidence. Currently, the fight against polypragmasy is one of the important tasks in rendering medical care to elderly and senile patients since it is a major risk factor of adverse drug reactions in this category of people.

Therefore, the manuscript submitted for review presents a current and clinically relevant problem. The manuscript is correctly structured in accordance with the requirements of scientific work. The introduction is a comprehensive and interesting introduction to the topic, the methodology is described in an accurate and meritorious way. The results are correctly and accurately described. However, the weakness of this manuscript is the discussion as it does not provide an explanation of the results and mostly describes known information from other publications. The discussion must be corrected - it should contain explanations of the results obtained and not resemble a review.

Author Response

Dear Academic Editor,

Dear Peer-Reviewers,

We are very thankful to you for the pertinent notes; we have carefully read the comments and have revised/ completed the manuscript accordingly. Our responses are given in a point-by-point manner below, as well, all the changes to the manuscript are highlighted in yellow.

We hope that in this new form, the manuscript will be suitable for publication in Medicina.

Reviewer 2

We would like to thank for your valuable comments which helped us improve this manuscript. All suggestions were taken into consideration and appropriate information as well as required corrections were provided. New/corrected parts are highlighted in yellow to facilitate the assessment of changes. We did our best to fulfil the expectations and we hope that you will be satisfied with our corrections.

At the present time, polypragmasy is a serious public health problem, as it is clinically manifested by a reduction in the effectiveness of pharmacotherapy, by the development of severe adverse drug reactions, and by a considerable increase in healthcare expenditures. The reason for the simultaneous prescription of multiple drugs may be comorbidity (multimorbidity), the availability of drugs, as well as treatment standards that contain recommendations for using combination therapy with more than 5 drugs for only one disease in some cases, the efficiency of which corresponds to a high level of evidence. Currently, the fight against polypragmasy is one of the important tasks in rendering medical care to elderly and senile patients since it is a major risk factor of adverse drug reactions in this category of people.

Therefore, the manuscript submitted for review presents a current and clinically relevant problem. The manuscript is correctly structured in accordance with the requirements of scientific work. The introduction is a comprehensive and interesting introduction to the topic, the methodology is described in an accurate and meritorious way. The results are correctly and accurately described. However, the weakness of this manuscript is the discussion as it does not provide an explanation of the results and mostly describes known information from other publications. The discussion must be corrected - it should contain explanations of the results obtained and not resemble a review.

Response: Thank you for your positive comments regarding our paper. We have taken into consideration your valuable suggestions and have amended the Discussions to explain the results obtained.

Thank you for taking time to review our paper and for your valuable suggestions. Overall, we believe that, thanks to you pertinent comments, the quality of the manuscript has been improved and its revised version warrants publication in Medicina.

Round 2

Reviewer 2 Report

The manuscript can be published in present version.